# MiR-199a-5p-Regulated SMARCA4 Promotes Oral Squamous Cell Carcinoma Tumorigenesis

**DOI:** 10.3390/ijms24054756

**Published:** 2023-03-01

**Authors:** Mingyan Xu, Junling Zhang, Xuemei Lu, Fan Liu, Songlin Shi, Xiaoling Deng

**Affiliations:** 1Department of Implantology, Stomatological Hospital of Xiamen Medical College & Xiamen Key Laboratory of Stomatological Disease Diagnosis and Treatment, Xiamen 361008, China; 2Engineering Research Center of Fujian University for Stomatological Biomaterials, Department of Stomatology, Xiamen Medical College, Xiamen 361023, China; 3Department of Basic Medical Science, School of Medicine, Xiamen University, Xiamen 361102, China

**Keywords:** SMARCA4, miR-199a-5p, oral squamous cell carcinoma, EMT

## Abstract

SWI/SNF related, matrix associated, actin-dependent regulator of chromatin, subfamily a, member 4 (SMARCA4, also known as BRG1), an ATPase subunit of the switch/sucrose non-fermentable (SWI/SNF) chromatin remodeling complex, plays an important regulatory role in many cytogenetic and cytological processes during cancer development. However, the biological function and mechanism of SMARCA4 in oral squamous cell carcinoma (OSCC) remain unclear. The present study aimed to investigate the role of SMARCA4 in OSCC and its potential mechanism. Using a tissue microarray, SMARCA4 expression was found to be highly upregulated in OSCC tissues. In addition, SMARCA4 upregulate expression led to increased migration and invasion of OSCC cells in vitro, as well as tumor growth and invasion in vivo. These events were associated with the promotion of epithelial–mesenchymal transition (EMT). Bioinformatic analysis and luciferase reporter assay confirmed that SMARCA4 is a target gene of microRNA miR-199a-5p. Further mechanistic studies showed that the miR-199a-5p regulated SMARCA4 can promote the invasion and metastasis of tumor cells through EMT. These findings indicate that the miR-199a-5p- SMARCA4 axis plays a role in tumorigenesis by promoting OSCC cell invasion and metastasis through EMT regulation. Our findings provide insights into the role of SMARCA4 in OSCC and the mechanism involved, which may have important implications for therapeutic purposes.

## 1. Introduction

Oral squamous cell carcinoma (OSCC) is the most common malignancy in the head and neck region, accounting for approximately 2% of new cancer cases and 1.8% of mortality worldwide in 2021 [1,2,3]. In the past two decades, numerous studies have shown that tumor metastasis is one of the main reasons the 5-year survival rate of OSCC is lower than 50% [4]. Therefore, understanding the molecular mechanism of metastasis of OSCC is of great significance to identifying new therapeutic intervention targets, developing new anticancer drugs, and improving the survival rate.

Epithelial–mesenchymal transition (EMT) is a major driving mechanism of the invasion and metastasis of tumor cells [5]. The hallmark of EMT is E-cadherin downregulation [6] and vimentin upregulation [7], resulting in loss of intercellular adhesion and increased cell motility [8]. SWI/SNF related, matrix associated, actin-dependent regulator of chromatin, subfamily a, member 4 (SMARCA4), the core ATPase subunit of the human SWI/SNF chromatin remodeling complex, regulates gene transcription through the regulation of chromatin structure [9]. It plays an essential role in a variety of cellular processes including differentiation, proliferation, and DNA repair [10,11,12]. Increasing evidence suggests that SMARCA4 plays an important role in tumorigenesis [13,14,15,16]. However, whether SMARCA4 acts as a tumor suppressor gene or oncogene remains controversial [17,18]. SMARCA4 was originally reported as a tumor suppressor gene due to its inactivating mutations or downregulation of its expression in several cancers and cancer cell lines [19,20,21,22,23]. However, SMARCA4 is also known to act as oncogene to promote tumorigenesis due to its high expression level found in certain cancers, including prostate cancer [24], breast cancer [25], liver cancer [26], and gastric cancer [27]. As an oncogene, several studies have shown that SMARCA4 promotes the occurrence and development of tumors by promoting the invasion and metastasis of tumor cells through the process of EMT [12,25,28,29]. In OSCC, no gene mutation but increased mRNA expression of SMARCA4 was detected in 62% of OSCC samples compared with normal controls [30]. However, the mechanism of SMARCA4 in promoting OSCC remains largely unknown.

MicroRNAs (miRNAs) are endogenous non-coding RNAs that modulate gene expression at the post-transcriptional level [31]. Alterations in miRNA expression have been implicated in many cancers, including OSCC [32,33,34]. Low expression of several miRNAs has been found in OSCC [35,36], among which downregulation of miR-199a-5p is thought to be closely related to the recurrence and progression of OSCC [35,36,37,38]. However, whether the upregulation of SMARCA4 in OSCC is due to reduced expression of miR-199a-5p in OSCC needs to be investigated.

The aim of the current study was to investigate the functional significance of SMARCA4 in promoting OSCC tumorigenesis both in vitro and in vivo, and the mechanisms involved.

## 2. Results

### 2.1. SMARCA4 Is Highly Expressed in OSCC

To examine the role of SMARCA4 in the progression of OSCC we performed tissue microarray analysis by IHC staining to assess the SMARCA4 expression levels in normal epithelial tissues and OSCC tissues. SMARCA4 was detected mainly in the nucleus of squamous carcinoma cells (weak staining in the region of squamous pearl formation) in about 75% of OSCC tissue samples, and normal oral mucosa specimens showed positivity mainly in sporadic cells of basal layers (Figure 1A). The proportion of SMARCA4-positive cells and the intensity of SMARCA4 protein immunostaining further confirmed that the SMARCA4 expression level in OSCC was significantly higher than that in normal tissues (Figure 1B). We also analyzed the SMARCA4 mRNA and protein expression levels in different OSCC cell lines, including SAS and CAL-27, in comparison to normal human epithelial cells HaCaT. The qRT-PCR and western blot analysis revealed increased expression levels of SMARCA4 mRNA and protein in both OSCC cell lines compared to that in HaCaT cells (Figure 1C,D). Therefore, the upregulation of SMARCA4 in the OSCC tissues is positively correlated with tumor progression, suggesting that SMARCA4 may play an oncogenic role in OSCC.

### 2.2. SMARCA4 Expression Is Associated with Tumor Invasion and Metastasis through EMT in OSCC

Since tumor invasion and metastasis are closely related to OSCC progression, we examined whether SMARCA4 is involved in OSCC cell migration and invasion. We first examined the effect of SMARCA4 overexpression on the invasion and metastasis of OSCC cells through a series of in vitro functional assays in OSCC cells (SAS and CAL-27) transiently transfected with a plasmid expressing SMARCA4 (pCMV5- SMARCA4). The expression of SMARCA4 increased five times for SAS cells and three times for CAL-27 cells after transfection with a pCMV5- SMARCA4 construct. The wound healing assay (Figure 2A,B) showed that overexpression of SMARCA4 markedly enhanced the mobility of SAS and CAL-27 cells. The Transwell migration assay showed that overexpression of SMARCA4 significantly promoted SAS and CAL-27 cell migration and invasion compared with their corresponding NC groups (Figure 2C,D).

The invasion and eventual metastasis of cancer cells require epithelial cells to undergo a process of de-differentiation known as EMT [39]. Loss of E-cadherin and upregulation of mesenchymal markers, such as vimentin, are thought to be key events in EMT. The immunofluorescence experiment showed that overexpression of SMARCA4 markedly attenuated E-cadherin expression in OSCC cells, whereas vimentin expression was upregulated (Figure 2E,F). The western blot analysis further confirmed these results (Figure 2G,H).

To further elucidate the effect of SMARCA4 expression on OSCC cell migration and invasion, we examined the effect of SMARCA4 knockdown on the invasion and metastasis of OSCC cells. The expression of SMARCA4 was knocked down about 70%–80% in both SAS and CAL-27 cells after transfection with the shRNA target SMARCA4 constructs. Wound healing assay of OSCC cells (SAS and CAL-27) transiently transfected with sh-NC, or sh-SMARCA4 (Figure 3A,B,C,D) showed that SMARCA4 knockdown markedly reduced the mobility of SAS and HSC3 cells. The Transwell migration assay (Figure 3E,F) additionally revealed that suppression of SMARCA4 expression markedly reduced OSCC cell migration and invasion, and these findings were confirmed by quantitative analysis (Figure 3G,H).

Overall, these results showed that overexpression of SMARCA4 significantly promoted OSCC cell migration and invasion, whereas inhibition of SMARCA4 reduced OSCC cell migration and invasion.

### 2.3. SMARCA4 Is a Target Gene of miR-199a-5p in OSCC

We investigated the underlying mechanism involved in SMARCA4 upregulation in OSCC. We used bioinformatic analysis tools and database, including ENCORI and MiRDB to predict a potential microRNA that may directly target SMARCA4. The results showed that SMARCA4 3′UTR has a conserved seed region for miR-199a-5p (Figure 4A). Therefore, we constructed plasmids containing the partial WT or MUT sequences of the 3′-UTR of SMARCA4 with miR-199a-5p target sites. The dual-luciferase assay results with these constructs showed that co-transfection of the WT plasmid with miR-199a-5p mimics significantly inhibited the relative luciferase activity compared with the co-transfection the WT plasmid with mimics NC, while co-transfection of the MUT plasmid with miR-199a-5p mimics blocked the inhibition of the luciferase activity by the WT plasmid, thus preventing the reduction of luminescence intensity (Figure 4B). In addition, significantly increased luciferase activity was observed in the group co-transfected with the WT plasmid and miR-199a-5p inhibitor, compared to the group co-transfected with the WT plasmid and inhibitor NC. In contrast, co-transfection of the MUT plasmid and miR-199a-5p inhibitor has no effect on luminescence intensity compared to co-transfection of the MUT plasmid with inhibitor NC (Figure 4C). These findings indicated that miR-199a-5p directly binds to specific sites in the 3′-UTR of SMARCA4. Thus, we examined the regulatory effect of miR-199a-5p on SMARCA4 expression in OSCC cells. SAS and CAL-27 cells were transfected with miR-199a-5p mimics or miR-199a-5p inhibitors, along with their corresponding NCs, and the transfection efficiency was assessed by qRT-PCR (Figure 4D,E). The results (Figure 4F–I) revealed that miR-199a-5p mimics markedly reduced SMARCA4 mRNA and protein expression, whereas inhibition of miR-199a-5p with its inhibitor significantly increased SMARCA4 mRNA and protein expression. Together, these results indicate that miR-199a-5p regulates the expression of SMARCA4 and SMARCA4 is a target gene of miR-199a-5p in OSCC cells. 

### 2.4. MiR-199a-5p-Regulated SMARCA4 Promotes OSCC Cell Migration and Invasion 

We firstly compared the expression levels of miR-199a-5p in normal epithelial cell line HaCaT and OSCC cells to examine the effect of miR-199a-5p on OSCC cell migration and invasion. The results showed that the miR-199a-5p expression level was significantly downregulated in OSCC cells, compared with HaCaT cells (Figure 5A). Next, the results of the wound healing assay showed that transfection with miR-199a-5p mimics dramatically reduced the migration capacity of SAS and CAL-27 cells within 6 h (Figure 5D,E), while miR-199a-5p inhibition restored the migration ability of SAS and CAL-27 cells within 6 h (Figure 5F,G). Additionally, the Transwell migration assay also revealed that SAS and CAL-27 cells transfected with miR-199a-5p mimics had less migratory and invasive ability than their corresponding mimics NC groups (Figure 6A,B). In contrast, miR-199a-5p inhibition increased the migration and invasion ability of SAS and CAL-27 cells compared with the corresponding inhibitor NC groups (Figure 6D,E). In addition, transfection with miR-199a-5p mimics significantly increased the E-cadherin protein expression level, whereas the vimentin protein level was decreased in OSCC cells (Figure 5B).

Having demonstrated that SMARCA4 is a direct target gene of miR-199a-5p in OSCC cells, we hypothesized that the role of SMARCA4 in OSCC cell migration and invasion is regulated by miR-199a-5p. To verify this hypothesis, we first investigated whether SMARCA4 overexpression can rescue the miR-199a-5p mimics-mediated inhibition of OSCC cell migration and invasion. To this end, we conducted various assays on SAS cells transfected with miR-199a-5p mimics alone, or co-transfected with miR-199a-5p mimics and SMARCA4. The results of the wound healing assays and Transwell assays demonstrated that the migration and invasion ability of SAS cells was restored by co-transfection of SMARCA4 with miR-199a-5p compared to cells transfected with miR-199a-5p only (Figure 7A,C), and these results were confirmed by quantitative analysis (Figure 7B,D). We also examined the effect of silencing SMARCA4 on the OSCC cell migration and invasion promoted by the inhibition of miR-199a-5p -. As expected, SMARCA4 inhibition successfully attenuated OSCC cell migration and invasion mediated by inhibition of miR-199a-5p (Figure 7E,G), and the results were also verified by quantitative analysis (Figure 7F,H). In summary, the above results suggest that the role of SMARCA4 in OSCC cell migration and invasion is regulated by miR-199a-5p.

### 2.5. Effect of SMARCA4 Knockdown on the Tumorigenesis of OSCC In Vivo

To further elucidate the tumorigenesis effect of SMARCA4 on OSCC in vivo, we examined the effect of SMARCA4 inhibition on tumor growth using a xenografts model in nude mice. As shown in Figure 8A, the xenograft tumors formed in mice of the sh-SMARCA4 group (with an average size of tumors about 100.84 ${\mathrm{mm}}^{3}$) were significantly smaller than those in the sh-NC group (with an average size of tumors about 374.6 ${\mathrm{mm}}^{3}$). In addition, the tumor growth rate (Figure 8B) and the tumor weight (Figure 8C) with SMARCA4 knocked down were significantly lower than those of the control group. The qRT-PCR analysis (Figure 8E) and immunohistochemistry staining analysis (Figure 8D) showed that the SMARCA4 level was significantly lower in the SMARCA4 knockdown group than that in the control group. The measurement of the changes in E-cadherin and vimentin levels in vivo revealed, as shown in Figure 8D, that E-cadherin was significantly upregulated, whereas vimentin was downregulated in the sh-SMARCA4 group compared with the sh-NC group (Figure 8D). These results further confirmed that SMARCA4 promotes the growth and metastasis of OSCC in vivo. Surprisingly, the measurement of the mRNA expression level of miR-199a-5p (Figure 8E) showed that the miR-199a-5p expression level was highly upregulated in the sh-SMARCA4 group.

## 3. Discussion

Previous studies have revealed that SMARCA4 plays a tumor-suppressive or oncogenic role in a context-dependent manner in various cancers. However, the function of SMARCA4 in OSCC remains unclear. This study investigated the potential role of SMARCA4 in OSCC tumorigenesis and its underlying regulatory mechanisms. Our results revealed that SMARCA4 is highly expressed in OSCC. Furthermore, SMARCA4 was found to be involved in the tumorigenesis of OSCC in vivo. We further found that miRNA 199a-5p directly targets SMARCA4 to modulate the EMT process, which may be an important link between SMARCA4 and OSCC cell migration and invasion. This study, for the first time, provides in vivo and in vitro evidence supporting the oncogenic role of SMARCA4 in OSCC tumorigenesis.

SMARCA4 is one of the most frequently mutated chromatin remodeling ATPases in cancer, and it manifests itself in a highly tumor-specific and tissue-specific manner [16,40]. Numerous studies have suggested that SMARCA4 is involved in tumorigenesis as a tumor suppressor, mainly through its loss-of-function mutations. However, mutation of the SMARCA4 gene was not detected in OSCC [30]. In fact, SMARCA4 mRNA level was found to be significantly upregulated in OSCC patients compared with that in matched normal controls [30]. Our results also showed that SMARCA4 is upregulated in both OSCC tissues and cell lines, which is consistent with a previous study [30]. Moreover, injection with SMARCA4-knockout SAS cells into nude mice significantly reduced tumor development. These findings strongly support the notion that SMARCA4 functions as an oncogene rather than a tumor suppressor gene in OSCC. In line with our finding, SMARCA4 has also been found to be highly upregulated and promote cancer development in other organ systems, such as prostate cancer [24], breast cancer [25], liver cancer [12], gastric cancer [41] and melanoma [29]. However, SMARCA4 has also been shown to be a tumor suppressor in various cancers, including lung cancer, colorectal cancer, and pancreatic cancer. Interestingly, SMARCA4 played both tumor-suppressive and oncogenic roles at distinct stages of pancreatic cancer formation [42]. Although its role in carcinogenesis remains controversial, it is generally recognized that SMARCA4 plays a dual role depending on the tissue type and cellular context.

SMARCA4 is involved in many cellular processes, some of which are associated with cancer development, such as differentiation, development, cell adhesion, growth control, metabolism, and DNA repair. Previous studies have shown that overexpression of SMARCA4 is involved in the development of various tumors by promoting tumor cell invasion and motility [12,25,28,29]. However, the specific function of SMARCA4 in OSCC remains unclear. In this study, although the efficiency of individual oral squamous cell carcinomas cells was different after transfection target SMARCA4 construct (overexpression or knockdown of SMARCA4), the transfection efficiency had no effect on OSCC cell migration and invasion ability. The different transfection efficiency might be due to experimental OSCC cells obtained from different OSCC individuals with their own genetic backgrounds. Therefore, the results of this study strongly support the notion that SMARCA4 is involved in the tumorigenesis of OSCC by promoting OSCC invasion and metastasis.

EMT is an essential process in cancer progression and metastasis [43,44,45]. During EMT, cells lose their plasticity, which facilitates cell migration and invasion. The typical biological changes in EMT are the decreased expression of the epithelial marker E-cadherin and the increased expression of mesenchymal markers, such as vimentin [46]. It has been reported that SMARCA4 synergizes with RUNX2 to promote EMT in colorectal cancer cells [47]. SMARCA4 promotes gastric cancer metastasis by suppressing E-cadherin expression and increasing vimentin expression [48]. Indeed, our study also showed that SMARCA4 regulates EMT-driven gene transcription to induce metastasis. Our results also showed that overexpression of SMARCA4 significantly decreased E-cadherin expression, but upregulated vimentin levels in OSCC cancer cell lines. These results were also confirmed in an in vivo nude mouse model. SMARCA4 depletion led to a marked decrease in the endogenous levels of E-cadherin that was coupled with an elevation of vimentin expression in the nude mouse model. It is therefore plausible that SMARCA4 promotes OSCC metastasis through EMT.

As a key component of the SWI/SNF chromatin remodeling complex, SMARCA4 plays a key role in regulating chromatin structure and gene transcription. However, the upstream regulation of the SMARCA4 gene expression remains largely elusive. The miRNAs represent a class of endogenous small non-coding RNAs that regulate gene expression at the posttranslational level and are important regulators of oncogenes and tumor suppressor genes in various tumors, including OSCC [33,49,50]. Several miRNAs have been reported to regulate SMARCA4 gene expression, including miR-199a-5p [51,52,53,54]. According to the previous study, miR-199a-5p is one of the most downregulated miRNAs in OSCC [35]. We, therefore, hypothesized that SMARCA4 upregulation may be partially due to the low expression of miR-199a-5p in OSCC. Through bioinformatic analysis and luciferase reporter assay, we confirmed that miR-199a-5p directly targets SMARCA4 in OSCC cell lines. Moreover, miR-199a-5p mimics significantly decreased SMARCA4expression levels, whereas miR-199a-5p inhibitor restored the SMARCA4 expression levels. These results indicate that SMARCA4 is a direct target gene of miR-199a-5p. Importantly, in this study, the expression of miR-199a-5p was low in OSCC cell lines with a high expression of SMARCA4. In addition, the miR-199a-5p expression level was significantly upregulated in the nude mouse model with SMARCA4 knockdown. These results suggested that there may be a negative regulatory relationship between SMARCA4 and miR-199a-5p, which may partially explain why SMARCA4 has been shown to be either overexpressed or downregulated in a variety of tumors. In these regards, SMARCA2 (also known as BRM), another catalytic subunit of the SWI/SNF complex, has been reported to form a double-negative feedback loop with miR-199a-5p in cancers [55]. Further research is warranted to corroborate the findings of the current study.

Previous studies have shown that miR-199a-5p plays a suppressive role in OSCC [35,37,38]. Accordingly, several mechanisms appear to be involved, including the inhibition of cancer cell migration and invasion by inhibiting EMT. Our study also showed that miR-199a-5p inhibited OSCC migration and invasion through inhibition of the EMT. These miR-199a-5p effects were in contrast to those of SMARCA4, further indicating the negative relationship between SMARCA4 and miR-199a-5p.

In conclusion, our study showed for the first time that SMARCA4 plays an oncogenic role in OSCC tumorigenesis. SMARCA4 was highly expressed in OSCC due in part to the low expression of miR-199a-5p. As a result, the high expression of SMARCA4 promoted OSCC cell migration and invasion through EMT, and the knockdown of SMARCA4 in vivo significantly suppressed OSCC metastasis (Figure 9). Therefore, the novel miR-199a-5p-regulated-SMARCA4 axis may serve as a novel potential diagnostic and anticancer therapeutic target in OSCC.

## 4. Materials and Methods

### 4.1. Cell Culture

The human keratinocyte cell line HaCaT was purchased from the Kunming Cell Bank of the Chinese Academy of Sciences (Kunming, China). The human OSCC cell line SAS was purchased from Japanese Collection of Research Bioresources (JCRB) cell bank (Tokyo, Japan). The CAL-27 cell line was obtained from the National Infrastructure of Cell Line Resources (NICR, Wuhan, China), All cells were cultured in Dulbecco’s modified Eagle’s medium (DMEM), and supplemented with penicillin (100 U/mL), glutamine (100 U/mL), streptomycin (100 U/mL), and 10% fetal bovine serum (FBS), in a humidified atmosphere containing 5% CO_2_.

### 4.2. Human Tissue Microarray

A human oral squamous cell carcinoma tissue microarray, containing 40 OSCC tissues and 8 normal oral mucosa tissues, was obtained from Alina Biological Technology Company (Xi’An, China). Immunohistochemistry (IHC) staining was conducted as previously described [56] using anti-SMARCA4 antibody (1:200, #49360S, Cell Signaling Technology, Danvers, MA, USA). The images were obtained by M8 Digital Scanning Microscope System (Precipoint, Freising, Germany). The images were examined independently by a pathologist. Optical density analysis of SMARCA4 expression in every piece of tissue (including 8 normal tissues and 40 OSCC tissues) was quantified by using Image Pro Plus version1 1.48 software. The SMARCA4 expressions were statistically analyzed using unpaired *t*-test.

### 4.3. Cell Transfection

Mimics and inhibitors of miR-199a-5p and corresponding negative control (NC) were synthesized by GenePharma Co., Ltd. (Shanghai, China). The oligonucleotide sequences are listed in Table 1. Transfections were performed using siRNA-mate (GenePharma Co, Ltd., Shanghai, China), according to the manufacturer’s instructions.

The human SMARCA4 expression construct was cloned into vector pCMV5-Flag (Addgene, Watertown, MA, USA), and an empty pCMV5-Flag vector was used as a negative control. Transfections were performed using Metafectene K4 (Biontex Laboratories GmbH, Munich, Germany) according to the manufacturer’s instructions. In order to construct a stable low-expression SMARCA4 lentivirus system, the SMARCA4-targeted short hairpin RNA (shRNA) sequences were synthesized and cloned into the lentiviral vector pLKO.1 (TRC Human SMARCA4 shRNA; Open Biosystems Inc., Huntsville, AL, USA). The recombinant plasmid pLKO.1-sh-SMARCA4 was verified by DNA sequencing. The pLKO.1-sh-SMARCA4, and packaging plasmids pVSV-G, and pHR (HanBio Therapeutics, Shanghai, China) were co-transfected into 293T cells using the TurboFect Transfection Reagent (Thermo Fisher Scientific Inc., Waltham, MA, USA) to produce pLKO.1-sh-SMARCA4 lentivirus. After transfection for 24 h, viral supernatants were collected to infect SAS or CAL-27 cells for 72 h. In all, 2 mg/mL puromycin (MedChemExpress, Monmouth Junction, NJ, USA) was used to select stable cells.

### 4.4. Protein Extraction and Western Blot Analysis

Cells were collected and lysed in radioimmunoprecipitation assay (RIPA) buffer containing a cocktail of protease inhibitors (MilliporeSigma, Burlington, MA, USA), and protein quantification was performed using the bicinchoninic acid (BCA) assay. Equal amounts of protein were separated by sodium dodecyl sulfate-polyacrylamide gel electrophoresis (SDS-PAGE) and then transferred to a polyvinylidene difluoride (PVDF) membrane (BioTraceTM NT; Pall Corporation, Ann Arbor, MI, USA). The membrane was blocked with non-fat milk for 1 h at room temperature followed by overnight incubation at 4 °C with anti-SMARCA4 antibody (1:1000, #49360S; Cell Signaling Technology Inc., Danvers, MA, USA), anti-E-Cadherin antibody (1:1000, #sc-8426; Santa Cruz Biotechnology Inc., Santa Cruz, CA, USA), anti-vimentin antibody (1:1000, #sc-373717; Santa Cruz Biotechnology Inc., Santa Cruz, CA, USA) and anti-GAPDH antibody (1:2000, #5174, Cell Signaling Technology Inc., Danvers, MA, USA). Membranes were then incubated with Anti-rabbit IgG antibody (1:2000, #7074; Cell Signaling Technology Inc., Danvers, MA, USA) or anti-mouse IgG antibody (1:2000, #7076, Cell Signaling Technology Inc., Danvers, MA, USA) at room temperature for 2 h. Membranes were eventually visualized with the ChemiDoc Touch Imaging System (Bio-Rad Laboratories, Hercules, CA, USA). Quantification of the resulting bands is achieved using densitometry software ImageJ. Results are normalized against reference protein GAPDH.

### 4.5. Quantitative Reverse Transcription Polymerase Chain Reaction (qRT-PCR)

Total RNA was extracted from cells or xenograft tumor tissue by using Trizol (Takara Bio, Shiga, Japan), and used to synthesize cDNA with the PrimeScript RT reagent Kit (Takara Bio, Shiga, Japan). Quantitative Real-Time polymerase chain reaction (qPCR) was performed using SuperReal PreMix Plus (SYBR Green, Tiangen Biotech Co., Ltd., Beijing, China) on an ABI 7500 Fast Real-Time PCR Detection system (Applied Biosystems, Foster City, CA, USA). Relative quantitation was performed using the 2^−ΔΔCt^ method [57], with GAPDH or U6 as the reference gene. Primer sequences were listed in Table 2.

### 4.6. Immunofluorescence

SAS cells transfected with the SMARCA4 overexpression plasmid were plated on glass slides for 24 h. Afterwards, the cell slides were rinsed with phosphate-buffered saline (PBS) and fixed in 4% paraformaldehyde for 30 min, followed by permeabilization with 0.2% Triton X-100 at room temperature. To avoid non-specific binding, the slides were then incubated with 5% goat serum for 1 h. Primary anti-E-cadherin antibody (1:300, #sc-8426; Santa Cruz Biotechnology Inc., Santa Cruz, CA, USA) and anti-vimentin antibody (1:300, #sc-373717, Santa Cruz Biotechnology Inc., Santa Cruz, CA, USA) were incubated at 4 °C overnight. The cell slides were next incubated with the corresponding secondary antibody (fluorescein isothiocyanate (FITC)-conjugated goat-anti-rabbit for E-cadherin, or DyLight-594-conjugated rabbit-anti-mouse for vimentin) in the dark for 1 h at room temperature. Immunofluorescence images were eventually acquired by confocal laser scanning microscopy using an Olympus Multi-Photon Laser Scanning Microscope (Olympus Corporation, Tokyo, Japan).

### 4.7. Bioinformatics Analysis and Dual-Luciferase Reporter Gene Assay

Various miRNA target prediction tools, including the Biosynthetic websites TargetScan (Accessed on 11 January 2021. Available online: https://www.targetscan.org/vert_80/), Starbase (Accessed on 11 January 2021. Available online: https://starbase.sysu.edu.cn/), and MiRDB (Accessed on 8 February 2021. Available online: http://mirdb.org/), were used to predict the binding sites between SMARCA4 and miR-199a-5p. The 3′UTR fragment of SMARCA4 targeted by miR-199a-5p and the mutated sequence were inserted into the reporter vector pmirGLO (Public Protein/Plasmid Library, PPL, NanJing, China) to construct the wild-type (WT) SMARCA4-3′UTR vector and mutated-type (MUT) SMARCA4-3′UTR vector, respectively. For the luciferase reporter assay, 293T cells were transfected with WT-SMARCA4-3′-UTR and miR-199a-5p mimics, MUT-SMARCA4- 3′-UTR and miR-199a-5p mimics, WT-SMARCA4-3′-UTR and miR-199a-5p inhibitor, or MUT-SMARCA4-3′-UTR and miR-199a-5p using METAFECTENE K4 (Biontex Laboratories GmbH, Munich, Germany). Dual-luciferase activity was detected 24 h later using the GloMax®-Multi+ Detection System (Promega, Madison, WI, USA), normalizing the reference of firefly luciferase to Renilla luciferase.

### 4.8. Cell-Based Assays

#### 4.8.1. Wound Healing Assay

Cells (5 × 105) were plated in 12-well plates. After 24 h when the cells were confluent, a micropipette tip was used to create a scratch wound in the cell monolayer in the center of each well. At the indicated time points (6 h for SAS cells, and 30 h for CAL-27 cells), the migration status was assessed by measuring the movement of cells into the scratched wound.

#### 4.8.2. Transwell Migration and Invasion Assays

Migration and invasion assays were performed using a 24-well plate Transwell assay system (Corning Inc. Cat#3422, Corning, NY, USA). Cells (1 × 105 for SAS cells or 2 × 105 for Cal-27 cells) in 200 µL of serum-free medium were resuspended and placed into the upper chamber for migration or 150mg Matrigel-coated chamber (BD Biosciences, San Jose, CA, USA) for invasion, and 600 µL of complete medium was added in the lower chamber as chemoattractant. After incubation for the indicated times (24 h for SAS cells, 48 h for CAL-27 cells), cells adhering to the surface of the membrane in lower chambers were fixed in 4% paraformaldehyde and stained with 0.1% crystal violet. Images of SAS and CAL-27 cells in migration and invasion Transwell assays were taken from six randomly selected fields under an IX51 Olympus microscope (Olympus Corporation, Tokyo, Japan). Quantification of the migrated cell number is achieved using densitometry software ImageJ. All experiments were repeated at least three times. The results were representative of at least three independent experiments.

### 4.9. Tumorigenicity Assays in Nude Mice

Ten 4–5 week-old (18–20 g) male nude mice were purchased from Vital River Laboratory Animal Technology (Beijing, China). The mice were randomly divided into two groups (5 mice/group). To establish a xenograft model, SAS cells (3 × 106, 100 µL) stably transfected with sh-SMARCA4 lentivirus or sh-NC were resuspended in PBS and injected into the right upper limb of each nude mouse. Tumor volumes and mouse weight were recorded every 4 days using scale. Two weeks later, nude mice were sacrificed using high concentrations of carbon dioxide, and tumor tissues were collected for subsequent analysis. Tumor tissues of nude mice embedded in paraffin were dewaxed in xylene and rehydrated in a graded alcohol series. The tissue morphology was observed by using HE staining. Immunohistochemistry (IHC) was performed as previously described [56] by using antibodies of anti-SMARCA4(1:200, #49360S, Cell Signaling Technology, Danvers, MA, USA), anti-E-cadherin (1:200, #sc-8426, Santa Cruz Biotechnology, Santa Cruz, CA, USA) and anti-Vimentin (1:200, #sc-373717, Santa Cruz Biotechnology, Santa Cruz, CA, USA). The images were obtained by microscope (IX51, Olympus America, Redmond, WA, USA). Optical density analysis of protein expression was conducted with Image Pro Plus. The images were also examined and scored independently by a pathologist. The animal experiments were approved by the Institutional Animal Experiment Committee of Xiamen University (XMULAC20200160). 

### 4.10. Statistical Analysis

All experiments were independently repeated three times for error analysis. Statistical results are presented as the mean and standard deviation (mean ± SD) or *p*-value. The difference between the two sets of data was analyzed using a *t*-test for paired data, while one-way analysis of variance (ANOVA) was used for more than two sets of data. A *p*-value < 0.05 was considered to be statistically significant, and the results were statistically analyzed and plotted using the Graph Pad Prism 7.0 software (GraphPad Software Inc., San Diego, CA, USA).

## Figures and Tables

**Figure 1 ijms-24-04756-f001:**
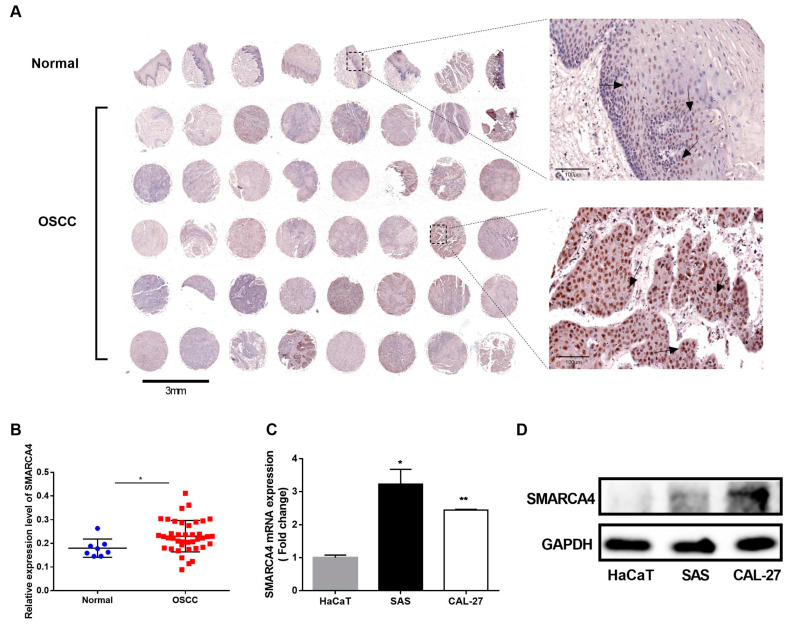
The expression of SMARCA4 is upregulated in OSCC. (**A**,**B**) A tissue microarray of human OSCC and normal tissues was analyzed by IHC staining using the Image ProPlus software (version1 1.48). (**C**) Comparison of SMARCA4 mRNA expression levels in OSCC cell lines (SAS, CAL-27) with that in HaCat cells by qRT-PCR analysis. Data are expressed as the mean ± SEM from triplicates experiments. * *p* < 0.05, ** *p* < 0.01. (**D**) Comparison of SMARCA4 protein expression levels in OSCC cell lines (SAS, CAL-27) with that in HaCat cells by western blot analysis.

**Figure 2 ijms-24-04756-f002:**
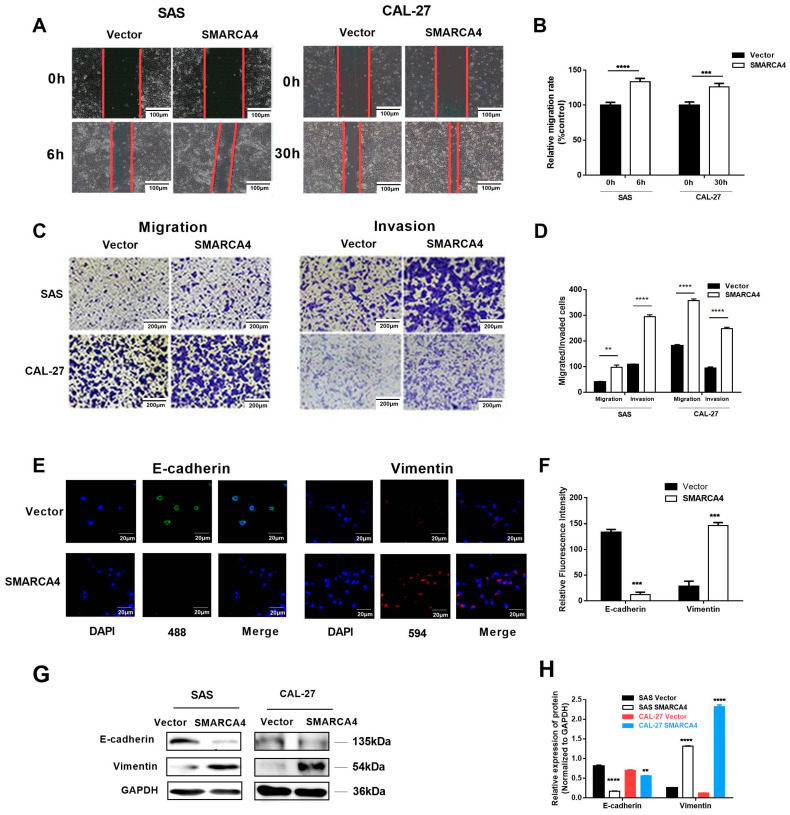
SMARCA4 promotes OSCC cell migration and invasion. (**A**,**B**) The effect of SMARCA4 overexpression on OSCC cell migration. SAS or CAL-27 cells were transfected with SMARCA4 overexpression plasmid or control vector for the indicated times (6 h for SAS, 30 h for CAL-27), and cell migration was assessed by the wound healing assay. Scale bar: 100 µm (**A**). Quantitative analysis was performed using the Image J version 1.48 software (**B**). (**C**,**D**) The effect of SMARCA4 overexpression on OSCC cell migration and invasion. SAS or CAL-27 cells were transfected with SMARCA4 overexpression plasmid or control vector for the indicated times (24 h for SAS, 48 h for CAL-27). Cell migration and invasion were assessed by Transwell migration and invasion assays. Scale bar: 200 µm (**C**). Quantitative analysis was performed using the Image J software (**D**). (**E**,**F**) Representative images of E-cadherin and vimentin expression in SAS cells detected by immunofluorescence staining after SAS cells were transfected with SMARCA4 overexpression plasmid or control vector for 48 h (Scale bar: 20 µm, green represents E-cadherin, red represents vimentin, and nucleus is shown in blue by DAPI staining). (**G**) The effect of SMARCA4 overexpression on OSCC cell E-cadherin and vimentin protein expression levels. SAS and CAL-27 cells were transfected with SMARCA4 overexpression plasmid or control vector for 48 h, and the E-cadherin or vimentin protein expression levels were measured by Western blot analysis. (**H**) Quantitative analysis was performed using the Image J software. Data are expressed as the mean ± SEM from triplicate experiments. ** *p* < 0.01, *** *p* < 0.001, **** *p* < 0.0001 vs control vector group.

**Figure 3 ijms-24-04756-f003:**
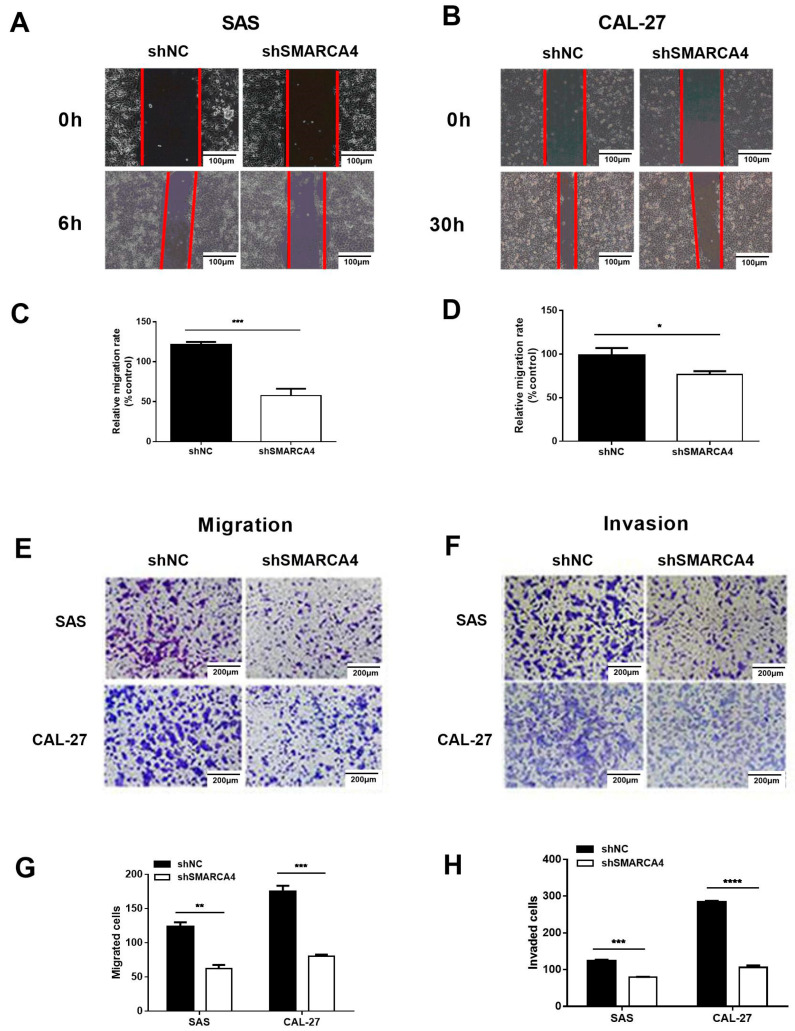
Knockdown of SMARCA4 inhibits OSCC cell migration and invasion. (**A**–**D**) The effect of SMARCA4 inhibition on OSCC cell migration. SAS or CAL-27 cells were transfected with sh-SMARCA4 or sh-NC for the indicated times (6 h for SAS, 30 h for CAL-27), and cell migration was assessed by the wound healing assay. Scale bar: 100 µm (**A**,**B**). Quantitative analysis was performed using the Image J software (**C**,**D**). (**E**–**H**) The effect of SMARCA4 inhibition on OSCC cell migration and invasion. SAS or CAL-27 cells were transfected with sh-SMARCA4 or sh-NC for the indicated times (24 h for SAS, 48 h for CAL-27). Cell migration (**E**) and invasion (**F**) were assessed by Transwell migration and invasion assays. Scale bar: 200 µm. Quantitative analysis was performed using the Image J software (**G**,**H**). Data are expressed as the mean ± SEM from triplicate experiments. * *p* < 0.05, ** *p* < 0.01, *** *p* < 0.001, **** *p* <0.0001 vs shNC group.

**Figure 4 ijms-24-04756-f004:**
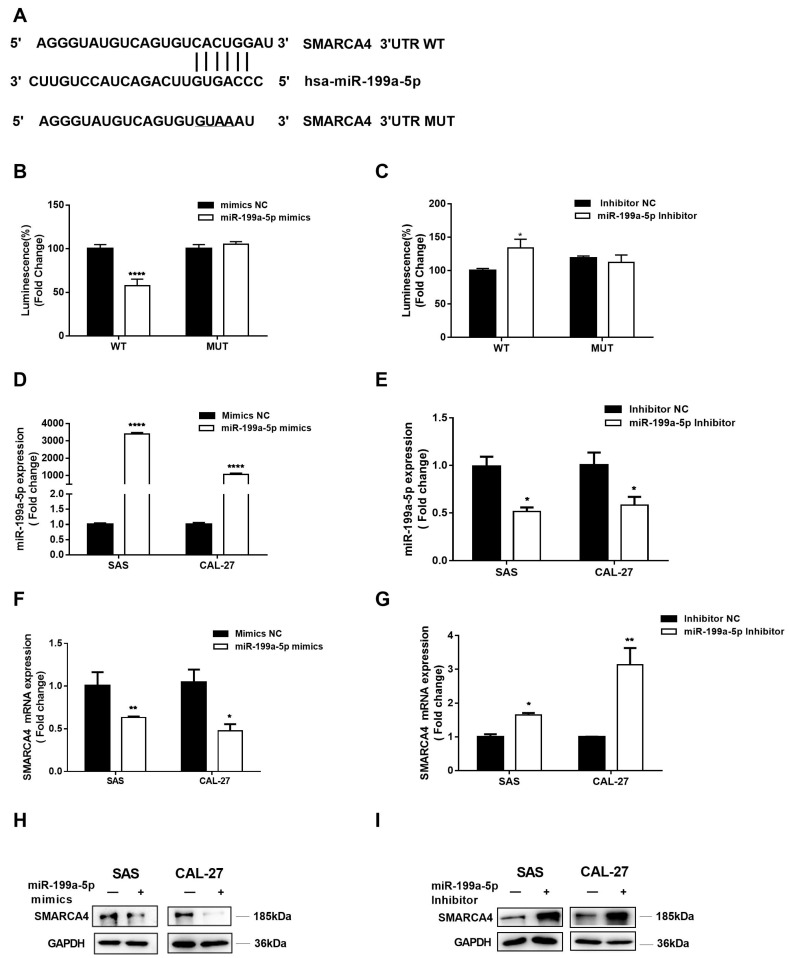
SMARCA4 is a target gene of miR-199a-5p in OSCC cells. (**A**) The alignment between miR-199a-5p and WT or MUT SMARCA4 3′UTR region. (**B**,**C**) SMARCA4 3′UTR containing a miR-199a-5p WT or a MUT target site was cloned into the luciferase reporter vector. These constructs were co-transfected with miR-199a-5p mimics, mimics NC, miR-199a-5p inhibitor, or inhibitor NC into 293T cells. Luciferase activity was measured after 48 h. (**D**,**E**) The transfection efficiency of miR-199a-5p mimics, mimics NC, miR-199a-5p inhibitor or inhibitor NC in OSCC cells. SAS or CAL-27 were transfected with miR-199a-5p mimics, mimics NC, miR-199a-5p inhibitor, or inhibitor NC for 48 h. The expression level of miR-199a-5p in OSCC cells was measured by qRT-PCR. (**F**,**G**) The effect of miR-199a-5p on SMARCA4 mRNA expression level in OSCC cells. SAS or CAL-27 cells were transfected with miR-199a-5p mimics (**F**) or miR-199a-5p inhibitors for 48 h (**G**). The mRNA expression level of SMARCA4 in OSCC cells was measured by qRT-PCR. (**H**,**I**) The effect of miR-199a-5p on SMARCA4 protein expression level in OSCC cells. SAS or CAL-27 cells were transfected with miR-199a-5p mimics (**H**) or miR-199a-5p inhibitors for 48 h (**I**). SMARCA4 protein level was measured by Western blot analysis. GAPDH was used as a loading control. Data are expressed as the mean ± SEM from triplicate experiments. * *p* < 0.05, ** *p* < 0.01, **** *p* < 0.001, comparison with the corresponding control.

**Figure 5 ijms-24-04756-f005:**
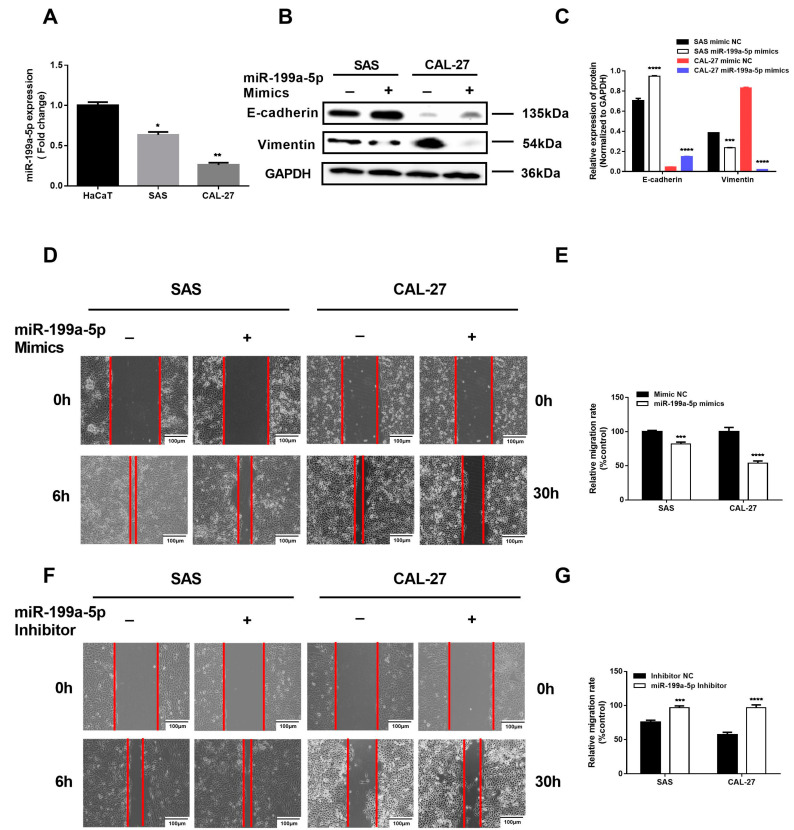
Effect of miR-199a-5p on OSCC cells EMT and migration. (**A**) Comparison of miR-199a-5p mRNA levels in SAS, CAL-27, and HaCaT. The miR-199a-5p mRNA level was detected by qRT-PCR. (**B**) The effect of miR-199a-5p mimics on OSCC cell E-cadherin and vimentin protein expression levels. SAS and CAL-27 cells were transfected with miR-199a-5p mimics 48h, and the E-cadherin or vimentin protein expression levels were measured by Western blot analysis. (**C**) The protein bands were analyzed by Image J. (**D**–**G**) The effect of miR-199a-5p mimics or miR-199a-5p inhibitor on OSCC cell migration. SAS cells or CAL-27 cells were transfected with miR-199a-5p mimics or miR-199a-5p inhibitor for the indicated times (6 h for SAS, 30 h for CAL-27), and cell migration was measured by the wound healing assay. Scale bar: 100 µm (**D**,**F**). Quantitative analysis was performed using the Image J software (**E**,**G**). Data are expressed as the mean ± SEM from triplicate experiments. * *p* < 0.05, ** *p* < 0.01, *** *p* < 0.001, **** *p* < 0.0001.

**Figure 6 ijms-24-04756-f006:**
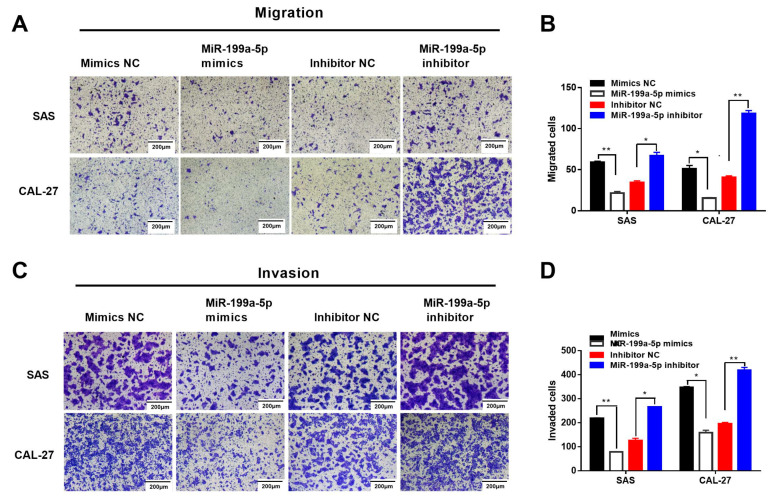
MiR-199a-5p inhibits OSCC cell migration and invasion. The effect of miR-199a-5p mimics or miR-199a-5p inhibitors on OSCC cell migration and invasion. SAS or CAL-27 cells were transfected with miR-199a-5p mimics or miR-199a-5p inhibitors for the indicated times (24 h for SAS, 48 h for CAL-27). Cell migration (**A**) and invasion (**C**) were assessed by Transwell migration and invasion assays. Scale bar: 200 µm. Quantitative analysis was performed using the Image J software (**B**,**D**). Data are expressed as the mean ± SEM from triplicate experiments. * *p* < 0.05, ** *p* < 0.01.

**Figure 7 ijms-24-04756-f007:**
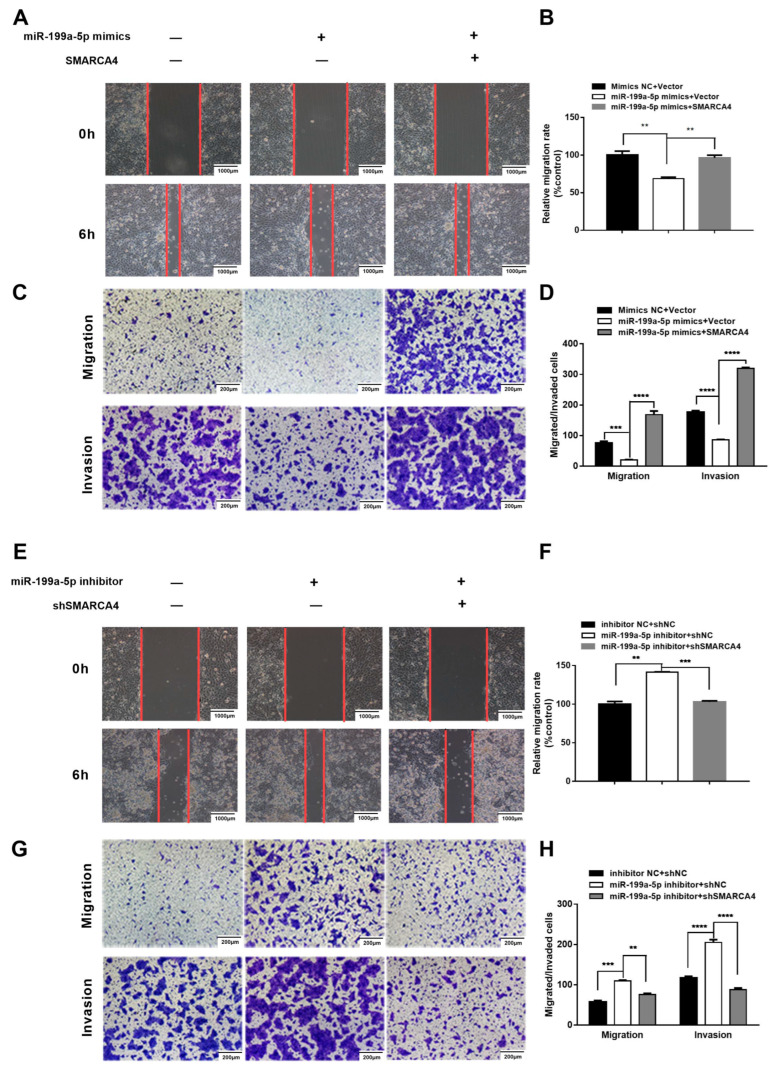
miR-199a-5p inhibits OSCC cell migration and invasion through the downregulation of SMARCA4. (**A**–**D**) The effect of SMARCA4 overexpression on miR-199a-5p mimics-induced OSCC cell migration and invasion inhibition. SAS cells were transfected with miR-199a-5p mimics, or miR-199a-5p mimics and SMARCA4 overexpression plasmid. Cell migration was measured by both the wound healing assay (**A**) and Transwell migration assay (**C**), and cell invasion was measured by the Transwell invasion assay (**C**). Quantitative analysis was performed by the Image J software (**B**,**D**). (**E**–**H**) The effect of SMARCA4 inhibition on miR-199a-5p inhibitor-induced OSCC cell migration and invasion. SAS cells were transfected with miR-199a-5p inhibitor, or miR-199a-5p inhibitor and sh-SMARCA4. Cell migration was measured by both the wound healing assay (**E**) and Transwell migration assay (**G**), and cell invasion was measured by the Transwell invasion assay (**G**). Quantitative analysis was performed by the Image J software (**F**,**H**). The scale bar is shown in lower-right corner. Data are expressed as the mean ± SEM from triplicate experiments. ** *p* < 0.01, *** *p* < 0.001, **** *p* < 0.0001, comparison with the corresponding controls.

**Figure 8 ijms-24-04756-f008:**
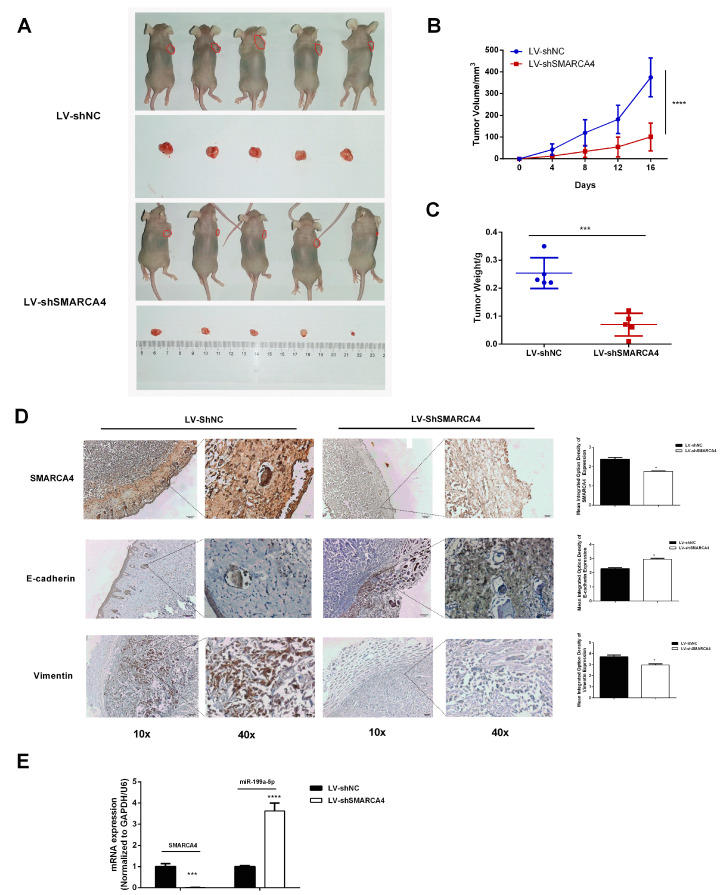
Knockdown of SMARCA4 suppresses OSCC tumorigenesis associated with upregulated miR199-a-5p in vivo. (**A**) Xenograft nude mice were injected with SAS cells stably transfected with sh-SMARCA4 or sh-NC. After 2 weeks, the mice were sacrificed, and the tumors were collected. The tumor size was measured using a caliper. (*n* = 5, each group). (**B**) Comparison of tumor growth rate between the sh-SMARCA4 group and sh-NC group. (**C**) Comparison of tumor weight between the sh-SMARCA4 group and sh-NC group. (**D**) Representative IHC staining images of SMARCA4, E-cadherin, and vimentin protein in tumor tissues from the sh-SMARCA4group or sh-NC group (scare bars: 10×: 100 µm, 40×: 20 µm). (**E**) Comparison of SMARCA4 or miR199-a-5p mRNA expression levels in tumor tissues from the sh-SMARCA4 group and sh-NC group. * *p* < 0.05, *** *p* < 0.001, **** *p* < 0.0001, comparison with the sh-NC group.

**Figure 9 ijms-24-04756-f009:**
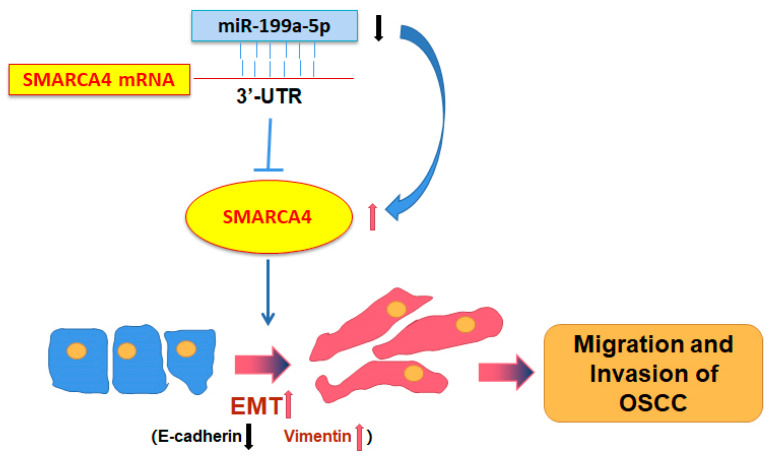
The proposed model for SMARCA4-promoted metastasis in OSCC. The miR-199a-5p -SMARCA4 axis regulates OSCC cell migration and invasion through EMT and plays a critical role in OSCC metastasis.

**Table 1 ijms-24-04756-t001:** Oligonucleotide Sequences of microRNA.

microRNA	Oligonucleotide Sequences (5′ > 3′)
hsa-miR-199a-5p mimics	CCCAGUGUUCAGACUACCUGUUC
	ACAGGUAGUCUGAACACUGGGUU
mimics NC	UUCUCCGAACGUGUCACGUTT
	ACGUGACACGUUCGGAGAATT
hsa-miR-199a-5p inhibitor	GAACAGGUAGUCUGAACACUGGG
inhibitor NC	CAGUACUUUUGUGUAGUACAA

**Table 2 ijms-24-04756-t002:** Sequences of oligonucleotide primers used in qRT-PCR.

Target	Forward Primer (5′ > 3′)	Reverse Primer (5′ > 3′)
has-miR-199a-5p	CGCGCCCAGTGTTCAGACTAC	AGTGCAGGGTCCGAGGTATT
U6	CTCGCTTCGGCAGCACA	AACGCTTCACGAATTTGCGT
SMARCA4	AGTGCTGCTGTTCTGCCAAAT	GGCTGGTTGAAGGTTTTCAG
GAPDH	GGAGCGAGATCCCTCCAAAAT	CTGGCCCGGAAGACATCTG

## Data Availability

The raw data supporting the conclusions of this article will be made available by the authors, without reservation.

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
