# Peer review of "MiR-199a-5p-Regulated SMARCA4 Promotes Oral Squamous Cell Carcinoma Tumorigenesis"

_ijms, 2023, doi:10.3390/ijms24054756_

Round 1

Reviewer 1 Report

Xu et al., investigated the role of Brg1 in oral squamous cell carcinoma (OSCC) and its underlying regulatory mechanism in tumorigenesis. The data collected by the authors revealed that OSCC is characterized by high expression levels of Brg1, and at the same time Brg1 is involved in the tumorigenesis of OSCC cells in vivo. The authors also found that miRNA 199a-5p directly targets Brg1 to modulate EMT process, that may represent the “missing” link between Brg1 and OSCC invasiveness. The study provides in vivo and in vitro evidence to support the oncogenic role of BRG1 along with OSCC tumorigenesis.

The article is interesting and well organized. Introduction and Discussion are fine. However, the Results are presented not always in a systematic and consistent manner.

Minor comments are found below:

Material and Methods:

- Section 2.2 Cell Transfection: I would suggest to put the oligonucleotide sequences in a table.

- Section 2.3 Protein extraction and Western blotting: Please, specify the meaning of BCA. It would be useful if the authors could provide more information about the quantification of the bands from the Western blotting analysis.

- Section 2.4 qRT-PCR: I would suggest to introduce a table for the primers.

- Section 2.7 Cell function assays: I would suggest to indicate this sub-section as “Cell-based assays”.

- Section 2.7.1: Where the cells were seeded? Which upper chamber? 96-, 48-, 24-multiwell format?

- Section 2.7.2: Please, specify the density of the cells plated in a 12-well plate format.

- Section 2.7.3: It is a little bit unclear how the authors performed the migration and invasion assay. I would suggest clarifying this section providing more information.

- Section 2.8: The use of nude mice in the study should be introduced in 2.1 “Cell culture and clinical samples”.

In 2.8 the authors should provide information about the tumors. Moreover, how these tumors were induced in the mice? Were all the tumors of the same origin and age? How old were the mice?...etc. What type of slides were used for the staining of the tumors? What type of microscope? Confocal?

- Section 2.9: As suggested above, I will move this sub-section in 2.1. Moreover, it will help to better understand the sections in Material & Methods where tumors are analyzed, as in section 2.8. The authors reported that tumor size and mouse weight were recorded every 4 days. If possible, it would be useful if the authors could provide more information (for example, how the animals were weighted?  …etc.)

Results:

General comment: All the Figures should be consistently labeled. So for example, I would suggest in all the Figures that contain representative images of WB, IF and IHC to move on the top, under the cell type indication, the presence and the absence of a treatment. As for example, in Fig. 4 F and G, miR-199a-p mimics should be moved on the top under SAS and CAL-27.

- Fig.1: I would suggest to move the “Normal” tissues on the top of the OSCC tumors. In this panel, I would also move the scale bar that otherwise covers the tissues. Magnification is barely visible within the two enlarged boxes. Scale bar and magnification should be also reported in the relative Figure legend.

- Section 3.1: Which software has been used to evaluate the expression of Brg1 by IHC? The authors have 8 Normal tissues vs 40 OSCC tumors. How has been processed the normalization? In addition, it is a little bit confusing what the authors mean when they introduce the “semi-quantitative statistical analysis…”.

- Section 3.2: What is the difference between SAS unmigrated vs migrated cells? And between CAL-27 unmigrated and migrated cells?  What does make these cells to acquire a migrating phenotype? The transfection with Brg1? This is unspecified in the text and Fig. 2B is confusing since it shows the same cell type but with different characteristics.

- Fig. 2C: how many folds Brg1 is overexpressed in SAS and CAL27 when compared to untransfected cells? Could the authors explain the different times of transfection?

- Fig. 2G: The authors have provided a panel of representative images 20 um. If possible, I would include a supplemental panel of representative images with higher magnification, in the range of 5-10 um to make more visible the different expression pattern.

- Fig. 2H: How the WB bands were quantified? How many biological replicates were used? How many times the WB was repeated? Please, provide more information.

- How was performed the migration and invasion quantitative analysis?

- Fig. 3: How did the authors evaluate the efficiency of transfection? What was the fold of knockdown? This piece of information must be included in the main manuscript.

- Fig. 4C: the authors stated that “In contrast, the significantly increased luciferase activity was observed by co-transfecting the plasmid containing the wild type and miR-199a-5p inhibitor, while the mutation-type plasmid blocked the induced luminescence intensities”. Where I can find this information? The panel contains WT and MUT with NC and miR-199a-5p inhibitors. The statement in very confusing. Please, modify, it and/or adjust the Figure accordingly.

- Fig. 5: This Figure reads very busy. I would suggest to split this panel in two separate Figures. One that would include the results from the scratch-wound healing assay and the other Figure that would contain the Invasion and Migration results.

- Fig. 5C and E: These two are different panel of images but the authors seemed to use the miR-199a-5p mimics and inhibitor for both. This is confusing. Please, modify it.

- Fig. 5D and F: The authors should have consistency when presenting the results. I would suggest using the legend as for example in Fig. 5H that is similar to Fig. 2 D and F, or as in Fig. 4B, C, D and E, etc.

- Fig. 5C, E, G, I: It would read easier if the authors move on the top miR-199a-5p mimics and inhibitor.

- Fig. 6B: Same comment of Fig. 5D and F. Please, adjust it accordingly.

- Fig. 6A and C: Same comment of Fig. 5C and E. Please, adjust it accordingly.

- Fig. 6D: I would move the Legend on the right since it overlaps with the graph. Please make a little bit bigger the graph itself.

- Section 3.5: Can the authors give an approximate average size of the tumors in the two groups?

- Fig7A: I would reorganize the image putting on the top the mice with the appropriate labeling and under each mice the panel with the relative tumors. I am not sure if the ruler can apply to both. The authors should adjust/modify this information.

- Fig. 7D: How the authors can claim that there is a clear overexpression of E-cadherin and down-regulation of vimentin? The quantification should be included to support this statement.

- Fig. 7E: Please, remove the labeling on the bottom of the graph since it appears already on the top right.

Reviewer 2 Report

 Xu et al., highlighted the role of BRG1 in promoting oral squamous cell carcinoma. The study adds to the domain knowledge.  

Few minor suggestions,

The authors mentioned bioinformatics analysis, but there is no related figure or supplement which can add to this. It is better to highlight this part as supplementary material.

Reviewer 3 Report

Paper submitted to MDPI IJMS titled: “BRG1 regulated by miR-199a-5p promotes oral squamous cell carcinoma carcinogenesis” is presentencing data on BRG1 expression and role for OSCC cells motility, invasion and tumor formation. The study is well planned and executed and organized with sufficient experimental evidence for publication. However prior publication couple of minor points should be addressed that include revised experiments and edits:

1.     In the Introduction please support the oncogenic function of Brg1 in different tumors with correct references.

2.     Please in one sentence add to the EMT transition and tumor progression, including growth factors dependence and markers for cancer cell changes of EMT in addition to -cadherin and vimentin.

3.     In the Methods section: 2.4 Quantitative real-time PCR (qRT-PCR), the formula should state 2 to the power -DDCt. Please list the sequence of the primers used for Brg1 gene in qRT-PCR.

4.     In the Methods section: 2.5 Immunofluorescence authors list two antibodies that are FITC conjugated , but in the Figure 2, two different colors are used for -Cadherin -green and for Vimentin-red. Please provide the correct fluorescent tags used for detection of these two proteins. It is not explained if DAPI or other counterstain for nucleus was used.

5.     In the Methods: 2.9 Tumorigenicity assay in nude mice authors neglect to say in what media the cancer cells were injected, giving only a volume.

6.     In the Results: 3.1 Brg1 is highly expressed in OSCC, authors should explain how the Brg1 protein was quantified from the IHC staining. In the Figure 1 please add arrows with indication for the staining of Brg1 high versus low (in normal tissues).

7.     In Figure 1C authors show increased Brg1 mRNA in OSCC cells as compared to normal, please add Western Blot to show the level of protein expression as well.

8.     In Figure 2B, authors show increased level of Brg1 mRNA of migratory cells as compared to Unmigrated cells. How the qPCR was normalized? Was it normalized to GAPDH level or relative to the cells seeded, or Non-migrated? Please explain in Figure legend.

9.     In the Results, 3.2 Brg1 is associated with tumor invasion and metastasis via EMT in OSCC, authors talk about testing SAS and HSC3 cells, however in Figure 3 (G,H), SAS with Cal-27 are shown. Please clarify and make consistent Results with Figure Legends and Final Figures.

10.  Please list other, in addition to Brg1, targets for miR-199a-5p, since miRs act through downregulating of other targets than Brg1.

11.  In Figure 4D, E, authors show expression level of miR-199a-5p, but neglect to show gene expression of Brg1. How overexpression of miR-199a-5p will affect stability of Brg1 mRNA?

12.  In Figure 7, in vivo experiment, tumor growth was conducted for 16 days, up to 400 mm^3 in the shNC and shBrg1. Although the differences between shNC and shBrg1 are significant the growth is still small to assess completely the tumor progression stage, since the tumors progress into metastatic sites at advanced size. Please clarify and add supplementary data on differences between shBrg1 and shNC at advanced disease stage in mice. In Figure 7D please show pictures of the full tumors stained with respective antibodies in order to assess E-cadherin and Vimentin protein distribution in the tumor versus surrounding tissue.

13.  Please carefully edit the manuscript and keep formatting consistent to present data in the highest quality.

Round 2

Reviewer 1 Report

The authors addressed major concerns and the manuscript has improved with the revision.